# OpenReview forum: "MetaAgent: Automatically Constructing Multi-Agent Systems Based on Finite State Machines"
_ICML.cc/2025/Conference — ICML 2025 poster_

### Official Review · Reviewer_6eD6 · 2025-03-07

**Overall Recommendation:** 3

**Summary:**

The paper proposes MetaAgent, an approach to automatically construct multi-agent systems using finite state machines (FSMs). Instead of hand-coding roles and workflows, MetaAgent uses a prompt-driven “Designer” to:

1. Identify which agents (roles) are needed to complete a family of tasks.
2. Build a finite state machine to represent the multi-agent collaboration flow, with possible traceback (going back to earlier states if errors are found) and null-transitions (staying in the current state for iterative refinement).
3. Optimize the initially generated FSM by merging redundant states before deploying it to real tasks.

**Claims And Evidence:**

Claims:
1. MetaAgent can generate multi-agent systems automatically (as opposed to requiring human-coded instructions).
2. Finite state machines provide superior flexibility over purely linear or orchestrator-based pipelines.
3. The generated multi-agent systems are robust and match or exceed the performance of both (a) domain-specific multi-agent frameworks designed by humans and (b) other auto-generated frameworks.

Evidence:

1. Each state is assigned one agent plus a separate condition-verifier mechanism that decides how to transition. This design is demonstrated in detail in examples (e.g., software development with states for requirements, coding, and testing).
2. On “Trivial Creative Writing” and “GPQA(Diamond)” question-answering, MetaAgent’s multi-agent system achieves higher accuracy (and coverage of correct answers) than other prompt-engineering baselines and the SPP auto-design approach.
3. MetaAgent produces a multi-agent system that nearly matches or surpasses specialized frameworks (e.g., DataInterpreter) and outperforms other auto-designed systems.
4. Outperforms MetaGPT (a well-known human-designed pipeline) and other auto-designed frameworks on a set of small projects (e.g., developing Snake, 2048, etc.), measured by functional checkpoints.

**Essential References Not Discussed:**

Within the discussion, the paper references many important works on multi-agent frameworks and tool-based LLMs (e.g., SPP, AutoAgents, Symbolic Learning, etc.). However, there are a few broader areas they only lightly mention or do not cite explicitly:
1. Formal methods for verifying the correctness of multi-agent systems: the authors mention automata (FSM) but do not elaborate on advanced formal verification techniques that might reduce error.
2. Additional references to “complex task orchestration” frameworks or thorough treatment of “self-play” from prior RL-based approaches might have offered more historical grounding.

**Experimental Designs Or Analyses:**

The paper uses:

1. Comparative performance with (a) multiple baseline prompt-engineering paradigms (e.g., Chain-of-Thought, Self-Refine, SPP) and (b) well-known multi-agent frameworks (MetaGPT, AutoAgents).
Multiple tasks that vary in complexity (short question-answering, creative writing, data science, and software dev). This broad coverage supports the authors’ claims of generality.
2. Cost analysis in tokens to highlight how the automatic design overhead compares to the cost of actually running tasks. They show that despite an up-front design cost (the LLM constructing the FSM), subsequent repeated usage can amortize that cost-effectively.

Strengths in design:

1. A variety of domain tasks confirm the FSM’s generality.
2. Clear success metrics (accuracy, coverage, or pass/fail of checkpoints).

Potential limitations:
1. The tasks tested are mostly small-scale and may not stress extremely large or complicated real-world scenarios (e.g., building production-grade software or enormous data pipelines).
2. While the paper reports improved performance on multiple tasks, seeing more ablation or failure modes in very complicated contexts would be interesting.

**Methods And Evaluation Criteria:**

Methods:

An LLM-based “Designer” first creates specialized agents—each with its own system prompt and tool access—relevant to the target domain. It then builds a finite state machine (FSM) by defining states, each tied to one agent, along with transition conditions and “listeners” to control information flow. The initially generated FSM may contain redundant states, so a second LLM-driven pass merges or removes overlapping roles. Finally, when the FSM is deployed, the system processes a user’s query state by state, allowing agents to backtrack or refine outputs until a final condition is met.

Evaluation Criteria:

1. Quantitative success rates on text-based tasks (e.g., how many questions are answered correctly in GPQA or how many writing tasks pass certain correctness checks).
2. Metrics on machine learning tasks such as F1-score, accuracy, or RMSE, aggregated into a “Normalized Performance Score (NPS).”

**Other Comments Or Suggestions:**

None

**Other Strengths And Weaknesses:**

Strengths

1. Unified framework: They argue how finite state machines unify or generalize popular multi-agent pipelines.
2. Traceback and Null-Transitions: This is an elegant way to add “debugging” loops and multiple tool calls within each state.
3. Experimental breadth: They tested text tasks, machine learning tasks, and code-generation tasks, demonstrating generality.

Weaknesses or Limitations

1. LLM Dependence: The entire pipeline’s success hinges on the correctness of the LLM used for designing the agents and transitions. The authors note performance degrades significantly when the LLM is replaced with a weaker model (e.g., GPT-3.5 vs. GPT-4).
2. Limited Depth on Formal Guarantees: While the authors position FSM as a powerful structure, they do not thoroughly discuss formal correctness or how reliably the condition verifiers handle ambiguous states in real, uncertain domains.
3. Cost–Benefit: Although the authors present cost analysis, real-world users might want more discussion or demonstrations of how well one can reuse a single “task-level” design across many similar sub-tasks without re-triggering the design step each time.
4. Minor: The fonts in Figures 1 and 2 are small.

**Questions For Authors:**

None

**Relation To Broader Scientific Literature:**

1. The authors mention synergy with code interpreters, search engines, etc., continuing a line of research that integrates LLMs with external APIs or tool frameworks.
2. The paper builds on the growing idea that large language models can “prompt themselves” (or use meta-prompts) to define roles, states, transitions, etc.

**Theoretical Claims:**

1. The authors argue that many standard multi-agent frameworks (linear pipelines, decentralized debate, orchestrator-based) can be seen as special or “constrained” forms of an FSM (with fewer transitions or no null-transitions).
2. By allowing cycles (traceback) and conditional transitions, an FSM can capture more sophisticated workflows and error-handling mechanisms—leading to better coverage of real-world complexities.

These claims are well-supported conceptually, showing how “traditional” multi-agent designs can be embedded in or derived from an FSM. That said, the paper does not delve deeply into formal language or automata theory beyond drawing analogies, so the theoretical underpinnings are relatively straightforward: it directly applies state machines to agent orchestration.

---

> ### Author Rebuttal · Authors · 2025-04-01
>
> Thanks for the reviewer’s appreciation of our finite state machine design as well as the thorough discussion and experiments.
>
> ## Re Reference Material:
> Thank you for your valuable feedback. To the best of our knowledge, our work is the first to introduce finite state machines for the automatic design of multi-agent systems. We appreciate the suggestion regarding formal verification techniques and RL-based approaches. In our revisions, we will expand the discussion to include relevant formal verification methods for multi-agent systems and provide a more comprehensive analysis of RL-based self-play approaches in the context of task orchestration.
>
> ## Re Wakeness 1:
> Firstly, our framework is orthogonal to the foundation model capabilities. A better foundation model leads to better performance. However, when using GPT3.5 as the designer, our method also shows good performance on the Machine Learning Task.
> Moreover, we admit this is a reasonable concern. But when using LLM as a designer of a Multi-Agent System and even the LLM Agent domain. Almost every existing work can not avoid the influence caused by the ability of the foundation model. Our framework shows good performance on many kinds of tasks empirically.
>
> ## Re Weakness 2:
> The condition verifier serves as a guider who utilizes the knowledge of the foundation designer model (by following state transition conditions) and helps the agent learn from feedback or evaluation results. So the thing is the same as Weakness 1: when the foundational model goes powerful, our FSM becomes more reliable in uncertain domains.
>
> ## Re Weakness 3
> As for the ‘reuse’, the benchmark on ML bench and Software already show the reuse ability of the designed FSM. Because the FSM use in these benchmarks is designed for general use in the task domain (eg. Software Development). When deployed, the FSM will handle different cases in the domain. Thus, the experiment is showing the ‘reuse’ of the FSM.

---

### Official Review · Reviewer_YubL · 2025-03-09

**Overall Recommendation:** 2

**Summary:**

This paper proposes a novel framework, MetaAgent, for the automatic generation of multi-agent systems based on finite state machines. The framework comprises three key steps: (1) Agent Design: The designer model defines the roles and tools for each agent according to task discriptions; (2) Finite State Machine Design: The designer decomposes the task into multiple states and assigns them to appropriate agents; and (3) FSM Optimization: Agents are dynamically merged based on role distinguishability and tool assignment. For experiments, this paper selects a range of text-based tasks (e.g., trivial creative writing and GPQA) and real-world coding tasks (e.g., ML benchmarks and software development tasks). The experimental results demonstrate the effectiveness and efficiency of the MetaAgent.

**Claims And Evidence:**

1. In Section 3.3, in the paragraph titled “Decentralized Debate as FSM,” this paper claims that those methods do not support null-transitions, which is presented as a key difference. This claim is not well-supported. The definition of null-transitions is not difficult, and they can be implemented through manual definition or prompt rewriting.

2. In Section 3.3, in the paragraph titled “Coordinate with Orchestrator as FSM,” the authors claim that those methods can be considered as FSMs. So, what is the main difference between them and MetaAgent?

**Essential References Not Discussed:**

None.

**Experimental Designs Or Analyses:**

The baseline comparisons are inconsistent. Tables 2, 3, and 4 use different baselines. Why is only SPP compared in Table 2? And why does the software development task, also a real-world coding task, have a different baseline setup compared to ML-Bench?

**Methods And Evaluation Criteria:**

1. Does predefining tools for a specific domain limit the ability to build automated workflows?

2. In section 3.4.3, with only the "merge" action available, is there a possibility that splitting might be necessary, for example, when agent1 and agent2 have similar roles in state 1 but different roles in state 2?

3. The construction of multi-agent workflows heavily relies on the designer’s abilities. In MetaAgent, the absence of hierarchical or tree-like decomposition places a high demand on the designer’s task decomposition skills.

**Other Comments Or Suggestions:**

1. The font size in Figure 1 is too small, resulting in poor readability.

2. In the second-to-last paragraph of section 3.4.3, which appendix is being referred to?

3. Sections 4.2 and 4.2.1 should be presented as parallel sections.

4. The explanation of the metrics would be clearer with the inclusion of formulas, such as for NPS.

5. In the second paragraph of section 4.4, “augment” should be corrected to “augments”.

6. Tables 9, 10, and 11 are disorganized.

**Other Strengths And Weaknesses:**

**Strengths**:

1. The authors’ introduction of Finite State Machines for modeling multi-agent workflows is insightful and inspiring for the field.

2. The authors thoroughly discuss different types of multi-agent frameworks and analyze the distinctions between MetaAgent and them.

**Weakness**:

1. This method heavily relies on the designer's capabilities for task decomposition and planning, and it does not provide any guarantee regarding the lower bound of the method's performance.

**Questions For Authors:**

In section 4.3, does the calculation of cost include the refinement process during the construction (as mentioned in section 3.4.3)?

**Relation To Broader Scientific Literature:**

This paper’s key contribution is using FSMs to build multi-agent workflows. While others focus on multi-agent framework construction, like automatic evolution, this work uniquely introduces FSMs and highlights their advantages over other approaches.

**Theoretical Claims:**

There are no theoretical claims.

---

> ### Author Rebuttal · Authors · 2025-04-01
>
> Thanks for the thoughtful feedback. We are encouraged that the reviewer agrees the Finite State Machine(FSM) is an inspiring method to the Multi-Agent System field and appreciates our theoretical analysis.
> # Re Weakness:
> Our framework is independent of the foundation model’s capabilities. Our method remains effective even with weaker models. For instance, when using GPT-3.5 as the designer, our approach still achieves strong results on Machine Learning benchmarks.
> Moreover, while concerns about the reliability of using an LLM as a designer in a Multi-Agent System, to the best of our knowledge, almost no existing work provides a provable guarantee. And Our framework shows good performance on many kinds of tasks empirically.
> # Re Claim 1:
> Simply modifying the prompt is insufficient to enhance the performance of the LLM debate structure on text-based tasks. This is because our finite-state machine (FSM) framework incorporates structural features such as Null-Transition and State Traceback, which cannot be replicated through prompting alone.
> To empirically validate this, we conducted an experiment comparing a purely prompt-modified LLM debate with the traditional one. Specifically, we rewrote the LLM debate prompt, instructed the model to refine its responses, and evaluated performance on GPQA (Diamond). The modified prompt achieved a score of 0.56, only marginally better than the traditional one (0.54). This result demonstrates that mere prompting cannot match the performance of our FSM-based approach—it is the FSM structure itself that drives the improvement.
> Theoretically, Null-Transition serves as a refinement mechanism, allowing agents to improve responses based on feedback. In contrast, Decentralized Debate follows a rigid sequence where agents present opinions without feedback opportunities, preventing output refinement. This limitation cannot be overcome through simple prompt modifications. However, introducing a condition verifier to refine responses and select speakers naturally transforms the structure into a finite-state machine.
> # Re Claim 2:
> Our MetaAgent Method represents the full realization of a Finite State Machine (FSM), as it incorporates a specialized condition verifier that enables Null-Transition and flexible state transitions, including state traceback.
> Section 3.3 demonstrates that existing Multi-Agent Systems can be viewed as limited FSMs; the "Coordinate with Orchestrator" approach, which has a shared verifier, suffers from centralized decision-making that becomes computationally burdensome as states increase. In contrast, FSM's decentralized architecture—with independent condition verifiers at each state—significantly improves scalability and adaptability.
> # Re Method1：
> We evaluate the MetaAgent Method on ML and Software Development tasks using two predefined tools: Python Code Interpreter and Search Engine.  Our experiments demonstrate that the designer LLM effectively selects and assigns tools to different agents.
> We also test FSM with an expanded tool pool; the designer maintains wise selections and consistent performance.
> # Re Method2:
> It is interesting to include the ‘splitting’ action in our method. However, practically, we find the designer LLM always tends to split the given task into trivial sub-tasks, causing a high failure rate because the chain of execution is too long. To fix this drawback, we design the merge action to help the designer LLM rethink and revise the FSM. The ‘splitting’ method is not practical because the initial version of the finite state machine itself tends to be very trivial.
> # Re Experiment 1
> We selected SPP since it is an auto-design multi-agent method.
> Based on your suggestion,  we have added new experiments for the software development task using AutoGen, Task Weaver, and Open Interpreter.
> This table shows the result:
> |Method|2048Game|SnakeGame|BrickBreakerGame|ExcelApp|WeatherApp|Avg|
> |------------------|-----------|------------|--------------------|-----------|-------------|------|
> |AutoGen|0.75|1|0|0|0|0.35|
> |Open Interpreter|0|0.5|0|0.25|0.25|0.2|
> |Task Weaver|0| 0.5|0|1|0|0.3|
> |MetaAgent|0.75|1|0.5|1|1|0.85|
>
> From the above table, we can observe that our method also outperforms existing baselines by a large margin. Note that we do not include  Data Interpreter since it is a frame specifically designed for Data Science.
> For other baselines in the text-based tasks, including direct prompt, CoT, CoT-SC, and Self-Refine. They are merely prompt-based, single LLM  methods instead of the multi-gent framework, and they do not support tool-using functionality. They perform poorly and are not a fair comparison with our method since we do not include them in the real-world coding task.
> # Re Comment 2:
> In appendix F
> # Re Comment 4:
> From DataInterpreter Paper.
> NPS =
> $\frac{1}{1 + s}$  (if s is smaller, the better )
> or
> $s$ (if s is bigger, the better ).
>
> # Re Question:
> Yes. It is included in the ‘design’ stage.’

---

### Official Review · Reviewer_rvy8 · 2025-03-13

**Overall Recommendation:** 3

**Summary:**

The paper introduces MetaAgent, a framework for automatically designing multi-agent systems using finite state machines (FSMs). The paper conceptualizes FSM within LLM agent design. The proposed method allows traceback ability to solve complex tasks. The paper also develops an optimization approach to merge the states for efficiency. Results on several text-based tasks demonstrate its performance compared with baselines.

## update after rebuttal

Thanks for the responses from the authors. I would recommend that the authors add the discussions on external data and foundation model capabilities in the revised version.

**Claims And Evidence:**

Supported: Tool usage and traceback are validated via ablation studies in Table 6. The overall performance is also evaluated on several benchmarks.
Not supported: The generalization ability is not tested across domains beyond text-based scenarios.

**Essential References Not Discussed:**

Several papers on flexible multi-agent collaboration architectures are not discussed. What are their relations with the FSM?

[1] Li, Guohao, et al. "Camel: Communicative agents for" mind" exploration of large language model society." Advances in Neural Information Processing Systems 36 (2023): 51991-52008.

[2] Guo, Xudong, et al. "Embodied LLM Agents Learn to Cooperate in Organized Teams." Language Gamification-NeurIPS 2024 Workshop.

[3] Chen, Weize, et al. "Agentverse: Facilitating multi-agent collaboration and exploring emergent behaviors in agents." arXiv preprint arXiv:2308.10848 2.4 (2023): 6.

**Experimental Designs Or Analyses:**

1. Though traceback helps with the performance, the paper seems not to report its potential influence such as delay in getting the results in practice.
2. The paper assumes LLMs can reliably decompose complex tasks, but does not evaluate failure cases (e.g., ambiguous user queries).

**Methods And Evaluation Criteria:**

1. The design of the method is clear and easy to follow.
2. Benchmark limitations: only text-based benchmarks are deployed, while real-world multi-agent tasks also include planning and decision-making tasks, such as VirtualHome (http://virtual-home.org/documentation/master/get_started/get_started.html).

**Other Comments Or Suggestions:**

The tables in Appendix C should be revised.

**Other Strengths And Weaknesses:**

Strengths:
1. State merging is a novel way to make multi-agent much more efficient.
2. FSM can potentially serve as a unified framework to develop more complex multi-agent system, which has been discussed in this paper.

Weaknesses:
1. Open-source models are not included and discussed.
2. The merging of states depends on one single LLM. The capability of this LLM will limit the performance.

**Questions For Authors:**

1. Is there any discussion on "the optimizing method does not need external data as well as numerous training steps"? If there is external data, will the optimization work better?
2. In ablation studies, the number of iterations is not clearly stated. The process of optimization is not included. Can you provide more information about this?

**Relation To Broader Scientific Literature:**

1. Multi-agent system: Extends prompt engineering to multi-agent coordination.
2. Automatic learning: Combining AutoML with agent workflow design.

**Theoretical Claims:**

N/A

---

> ### Author Rebuttal · Authors · 2025-04-01
>
> Thank the reviewer for appreciating the finite state machine as a unified framework of a Multi-Agent System.
> # Re Reference:
> CAMEL is a simple two-agent chat system resembling a Decentralized Debate structure (Session 3.3).
>
> AgentVerse employs two cooperation structures: Horizontal : a Linear System where decisions are summarized and Vertical:a Debate System where a solver and reviewer iteratively refine decisions until reaching consensus. These structures are also discussed in Session 3.3.
>
> The paper "Embodied LLM Agents Learn to Cooperate in Organized Teams" introduces a Communication-Action model, where agents communicate first and act later. However, this model has drawbacks: agents receive no immediate feedback during the Action Phase, delaying behavior refinement, and the leader agent can only issue instructions during the Communication Phase.
>
> In contrast, the FSM structure enables immediate feedback after each action via condition verifiers. Agents refine actions on the spot, determine state transitions, and communicate continuously, ensuring seamless interaction throughout the process.
>
> # Re Question 1:
> The proposed optimization method relies on a self-iteration procedure, where the system refines itself over time. While incorporating external data could enhance this process by allowing the designer agent to learn from additional information, our observations indicate that most failure cases in the initial version of the FSM are caused by overly trivial agent and state assignments. Therefore, the most effective approach is to prompt the designer agent to self-refine the FSM by merging trivial agents and states, eliminating the need for external data.
>
> Our FSM structure is designed to handle general open-ended, real-world tasks, where user inputs can vary widely. This variability makes it difficult to collect or synthesize high-quality test cases. As a result, relying on the self-iteration optimization method becomes not only effective but also the most practical approach for improving the FSM's performance in these complex, unpredictable scenarios.
> # Re Question2:
> As presented in the optimization method (Session 3.4.3):” This iteration continues until no further states can be merged and the state set stabilizes.” The ‘iteration’ in the ablation study means the optimization method. We will refine our description in the revision.
> # Re Evaluation:
> We have implemented a Virtual Home environment to the MetaAgent Framework and tested it with some housekeeping tasks. The following is a highlighted running log.
>
> **Agent Design**:  LivingRoomAgent, RestRoomAgent, KitchenAgent, and BedroomAgent.
>
>
> **State Design**(show state instructions):
> Take books from the living room, Clean rubbish in the living room, …, Take books from the bedroom
>
> **State Transition Example**:
> {
>                "from_state": "1",
>                "to_state": "2",
>                "condition": "If books are taken from the living room"
>  },
>
> When deployed, this multi-agent system can plan and submit actions to the environment in the specific format.   (example task: Clean the  bedroom)
>
> The Bedroom Agent first **plan**:
> Prioritized Cleaning Plan for the Bedroom:
> 1. **Organize Clothes:**
>   - Fold and organize the clothespiles (15 clothespiles).
>   - Hang the clothesshirts (3 clothesshirts) and clothespantss (5 clothespantss) in the closets.
> 2. **Clean Surfaces:**
>   - Wipe down the desk, nightstands, coffeetable, and bookshelf.
>   - Ensure the desk and nightstands are properly closed.
> …
>
> And then it **act**:
>
> <ACTION>
> {
>    "action": "Grab",
>    "object": "clothespile",
>    "object_id": 150
> }
> </ACTION>
> …
>
> Several test cases show that the FSM-based multi-agent system can also understand the Virtual Home environment and interact with it properly.
> # Re Weakness 1:
> When writing the paper, Open source models do not have enough intelligence level to achieve this complicated task of designing a Multi-Agent System. To invest the influence of the foundation model quality, we also apply GPT-3.5 as a weaker model in the ablation study, whose performance is lower than GPT-4o but is still comparable with other baselines.
> Recently, we also tried some powerful open-sourced models as designers (like deepseek-v3). It also worked well on our test cases.
> # Re Weakness 2:
> Firstly, our framework is orthogonal to the foundation model capabilities. A better foundation model leads to better performance. However, when using GPT3.5 as the designer (GPT-4o as executor), our method also shows good performance on the Machine Learning Task. (In ablation study, Table 5)
> Moreover, while concerns about the reliability of using an LLM as a designer in a Multi-Agent System (or even within the LLM Agent domain) are valid, to the best of our knowledge, almost no existing work in the area can avoid the influence caused by a weaker foundation model.

---

### Official Review · Reviewer_WYux · 2025-03-24

**Overall Recommendation:** 3

**Summary:**

This paper primarily discusses the automated construction of multi-agent systems. Its highlight is the introduction of the finite state machine (FSM) concept, incorporating null-transition states and state traceback into multi-agent systems. This allows the system to more flexibly address two issues: (1) when the current agent does not resolve a subtask as expected, and (2) when a downstream agent identifies problems with a previous agent, enabling traceback. Additionally, the paper outlines the relationships between three mainstream multi-agent approaches under this framework: Linear System, Decentralized Debate, and Coordinate with Orchestrator, explaining that they are all special cases of FSM. Based on the FSM idea, it Introduces the process of building and optimizing multi-agent systems. Its effectiveness was measured in scenarios such as Trivial Creative Writing, GPQA, and Coding. Furthermore, cost analysis and ablation experiments were conducted, demonstrating a comprehensive exploration.

**Claims And Evidence:**

The paper mentions that external data is not needed, but in the ablation experiment of "Reduce system redundancy through optimization", it shows that "... a few iterations are required to make the system more robust. . After testing the initial version of the multi-agent system on the pertinent test cases, the multi-agent system will be adapted in the aspect of agent and state design ...", which seems to be Inconsistente.

**Essential References Not Discussed:**

N/A

**Experimental Designs Or Analyses:**

In cost analysis, it is not clear here why the number of tokens used in this paper would be less. It stands to reason that additional verifiers would increase the use of tokens.

**Methods And Evaluation Criteria:**

For text-based tasks, the definition of metrics are provided, but it doesn't mention how to obtain them.

When evaluating software development task, the authors apply the "objective" checkpoints based methods. It would be better if the subjective evaluation metrics are reported together.

**Other Comments Or Suggestions:**

See questions.

**Other Strengths And Weaknesses:**

Main highlight is to use FSM to unify current multi-agent systems, and provide a way to automatically construct multi-agent systems.

**Questions For Authors:**

1. What will be the difference between a multi-agent system with single verifier and MetaAgent if the verifier knows the role of each agent?
2. How to get the metrics for text-based tasks
3. What is the exact meaning of "iterations" in the ablation study? Is it case-driven?
4. Why MetaAgent costs less with more verifier?

**Relation To Broader Scientific Literature:**

A framework to unify existing multi-agent systems, which is well-discussed in Session 3.3.

**Theoretical Claims:**

N/A

---

> ### Author Rebuttal · Authors · 2025-04-01
>
> Thanks for the reviewer’s effort and the appreciation in the discussion in Session 3.3. We believe the finite state machine has the potential to be a unified structure of the Multi-Agent System.
> # Re Claims1:
> Our optimization method is inherently self-iterative, meaning it does not rely on external training data. In the ablation study, the description highlights the motivation behind this design choice. Initially, we observed that the first version underperformed on test cases, which led us to develop this self-iteration approach as an optimization strategy. We will refine the description in the revision to improve clarity.
> # Re Question 1:
> This question explores why a Multi-Agent System, where agents are assigned different roles, can outperform a single LLM agent in certain tasks. Intuitively, assigning specific roles to agents activates their specialized knowledge related to those roles. Similarly, when all condition verification tasks are assigned to a single LLM, two main challenges arise. First, as the number of agents increases, a single condition verifier struggles to understand each agent’s situation. Second, dedicated condition verifiers, assigned to specific agents, can better capture and process the state-specific information, leading to more accurate and efficient verification.
> # Re Question 2:
>  GPQA evaluates the accuracy of multiple-choice questions, while Writing assigns a score based on keywords. Each keyword has a corresponding list that includes its various forms, and as long as the generated text contains at least one of these variants, it is considered valid for scoring.
> When designing the objective evaluation criteria for software development tasks, we also select several bad cases for subjective evaluation. Through this process, we gradually update the objective evaluation criteria to make it more reasonable.
> # Re Question3:
> No. It means the optimization method described in the method section. We will refine the description in the revision.
> # Re Experiment Metric also Question4:
> For a batch of tasks (10+), the cost of the MetaAgent architecture is lower than that of other frameworks. The primary reason is that the FSM is more general: it only requires a one-time design for the task domain, whereas other frameworks necessitate case-by-case design(eg, AutoAgents,SPP). As a result, MetaAgent incurs lower costs when handling a batch of tasks.

---

### Decision · Program_Chairs · 2025-05-01

**Decision:**

Accept (poster)

**Comment:**

The paper presents MetaAgent, a framework for automatically constructing multi-agent systems via finite state machines (FSM). The use of FSMs provides a unifying perspective over existing agent architectures and enables traceable, modular, and optimizable workflows. While the method’s reliance on a strong designer model is a valid concern, the authors address this through empirical evaluations and ablation studies across diverse tasks. The paper is well-executed and offers a novel and practical contribution, though some claims would benefit from broader real-world validations.